# An analysis of production efficiency in China's real estate industry based on a two-stage DEA model

Jiening Meng [1,2]*, Wei Bu[1]

**1** School of Economics and Management, Beijing Jiaotong University, Beijing, China, **2** School of Business, Henan University of Science and Technology, Luoyang, Henan, China

* 16113105@bjtu.edu.cn

## Abstract

To examine the resource utilization in different phases such as development and sales within China's real estate industry, this paper employs a two-stage Data Envelopment Analysis (DEA) model to measure the production efficiency of the real estate industry across 31 provinces, municipalities, and autonomous regions of China from 2014 to 2022. By examining both overall and phase-specific trends, the study utilizes a panel Tobit model to explore the factors affecting efficiency. Empirical results indicate that the leverage ratio of companies, per capita GDP of regions, and real estate regulatory policies significantly impact production efficiency. Further analysis of regional heterogeneity and its effect on production efficiency revealed that the per capita residential building area, which reflects the housing stock configuration in different regions, exhibits a significant single threshold effect. This not only objectively assesses the utilization of real estate resources in different areas but also delves deeper into the principal factors and their mechanisms affecting the production efficiency of the real estate industry, thus providing theoretical support and policy recommendations for effectively enhancing production efficiency.

**Data Availability Statement:** All relevant data are within the paper and its Supporting information files.

## 1 Introduction

Since the reform of the housing distribution system to a monetized allocation in 1998, China's real estate industry has entered a period of rapid development. To date, the real estate market in China has achieved a basic balance between supply and demand, yet regional and structural contradictions remain prominent [1], with real estate bubbles or inventory issues becoming significant challenges in various regions [2–4]. The development model of blindly expanding the scale of real estate investment is no longer appropriate for most areas. Therefore, how to develop real estate more effectively has become a topic of widespread concern across various sectors of society. Given the immobility of real estate in space, and in a context where factors such as population and capital are mobile, real estate markets in specific regions face unique supply and demand challenges. Hence, focusing on the regional attributes of the real estate market is crucial. Although many scholars have observed spatial differentiation in real estate

**Funding:** This article received data and funding support from the 2020 National Social Science Fund project "Research on Enhancing the Basic Capacity of China's Manufacturing Industry under Global Value Chain Reconstruction" (project number: 20BJY097). "The funders had no role in study design, data collection and analysis, decision to publish, or preparation of the manuscript".

**Competing interests:** The authors have declared that no competing interests exist.

prices [5–8], these prices do not accurately and comprehensively reflect the utilization of real estate resources in different regions. To overcome this dilemma, some scholars have started to study the production efficiency of the real estate industry in different regions and identify influencing factors [9, 10]. Existing research provides valuable references for the analysis presented in this paper, particularly in measuring production efficiency, estimating trends, and discussing influencing factors. However, most literature merely organizes thoughts on real estate production efficiency intuitively and lacks a discussion on the deep-seated reasons behind its constitution and changes, thus also lacking differentiated strategies when addressing various types of market issues.

To address these deficiencies, this paper sets forth two research objectives: 1. To use a two-stage DEA model to measure the production efficiency of the real estate industry across all 31 provinces, municipalities, and autonomous regions in China, accurately evaluating the resource utilization in different stages such as development, construction, and sales management; 2. Based on the overall, phase-specific, and regional characteristics of real estate industry production efficiency and considering both internal and external business environments, to explore its main influencing factors and mechanisms. This paper innovates by (1) emphasizing the regional attributes of the real estate market and deeply exploring the underlying reasons behind generally low production efficiency in the real estate industry from the socio-economic development status of the 31 provinces, municipalities, and autonomous regions, providing useful decision-making references for real estate developers and government managers; and (2) using a panel threshold effect model to test the impact of the existing configuration of real estate resources in various regions on production efficiency, offering more targeted suggestions for promoting regional economic development and optimizing public policy.

The rest of the paper is organized as follows: Section 2 reviews the institutional background and related literature. Section 3 describes the research methods and data sources. Section 4 provides quantitative analysis results and objectively assesses the utilization of real estate resources in various regions. Section 5 tests the impact of different factors on the production efficiency of the real estate industry. Section 6 concludes the paper and proposes targeted recommendations.

## 2 Institutional background and literature review

The development of China's real estate industry has been inherently linked to the regulatory measures governing the market. To address issues at different stages of the real estate market's evolution, regulatory policies have continuously evolved and academic research has progressively deepened. Initially, the aims of regulation were to "stabilize housing prices" and "stabilize the economy," choices made opportunistically. Over time, these aims have shifted towards establishing a housing system that incorporates multiple suppliers, diverse guarantee channels, and supports both renting and buying, as well as fostering a long-term mechanism for the stable and healthy development of the real estate market [11]. Regulatory tools have also evolved from initial adjustments of down payment ratios, loan interest rates, and business tax exemption periods to include more recent measures such as purchasing eligibility restrictions and sales limitations. A comprehensive real estate policy framework has been established that integrates monetary, fiscal, land, and administrative policies [12–14]. The regulatory approach has shifted from a one-size-fits-all demand control strategy to a more cautious and targeted "city-specific policy" approach [15, 16]. Given the current low overall production efficiency in China's real estate industry, analyzing the resource utilization in different segments such as production and sales, exploring related influencing factors, and proposing targeted regulatory strategies are critical for the sustained and healthy development of the industry. And this

requires an accurate understanding of the production efficiency in various regions as a prerequisite.

Academic research on real estate production efficiency varies by the subject of study: For real estate development companies: Studies, such as those by Atta Mills, E. F. E., Baafi, M. A., Liu, F., & Zeng, K., have measured the dynamic operational efficiency of listed Chinese real estate companies and analyzed their influencing factors [17, 18]. For national or regional real estate sectors: Research has been conducted on the production efficiency of the real estate industry along the Yangtze Economic Belt [19] and international real estate markets [20]. Research methods are divided into: Parametric analysis: This involves constructing a production function to describe the relationship between inputs and outputs of production units. The absolute production efficiency of the real estate industry is determined through regression analysis to solve for parameters within the function expression [21]. Non-parametric analysis: This method does not require a specific production function or the introduction of various behavioral assumptions, and calculates the relative production efficiency of the real estate industry through linear programming based on input-output relationships [22, 23]. Previous studies have not only objectively evaluated the production efficiency of the real estate industry but have also identified main influencing factors including regional economic development levels, residents' disposable incomes, the scale of the real estate industry, and regulatory policies [24]. However, there is a relative lack of research analyzing real estate production efficiency from the internal and external development environments of real estate development enterprises, and specifically across different stages and aspects of real estate development and sales. Therefore, building on existing research, this paper uses a two-stage DEA model to measure the production efficiency of the real estate industry in China's 31 provinces, municipalities, and autonomous regions from 2014 to 2022. It then sets these efficiency values as the dependent variables, with indicators reflecting the real estate production and sales conditions in each region as explanatory variables. This study constructs a panel Tobit model and a threshold effect model to test the specific effects of various factors on the production efficiency of the real estate industry.

## 3 Research methods and data description

### 3.1 Model selection

**3.1.1 Two-stage DEA model.** Data Envelopment Analysis (DEA), proposed by Charnes, Cooper, and Rhodes in 1978 [25], is commonly used to assess the production efficiency of multiple Decision Making Units (DMUs) that have multiple inputs and outputs. It utilizes mathematical programming and statistical data to establish a relatively efficient production frontier. DMUs are then projected onto this DEA frontier, and their efficiency is evaluated by measuring their deviation from the frontier. Given the lengthy development and construction cycles in real estate and the difficulty in achieving market clearance in sales, evaluating the production efficiency of the real estate industry across different regions requires a phased approach to examining resource utilization in both the development and sales stages. Consequently, this paper employs a two-stage DEA model to calculate the production efficiency of the real estate industry in various regions of China. The two-stage DEA model assumes there are $n$ DMUs, each with a production process that can be divided into two consecutive sub-processes, as illustrated in Fig 1. Each DMU has p types of inputs, $s$ types of intermediate outputs, and $q$ types of final outputs. Let $x_{ij}(i = 1, 2, . . ., p)$ denote the i-th type of input for the $j$-th DMU, $y_{rj}(r = 1, 2, . . ., q)$ denote the $r$-th type of output for the $j$-th DMU, $z_{mj}(m = 1, 2, . . ., s)$ denote the $m$-th type of intermediate product for the j-th DMU, which serves as both the m-th output of Stage 1 and the $m$-th input of Stage 2; and

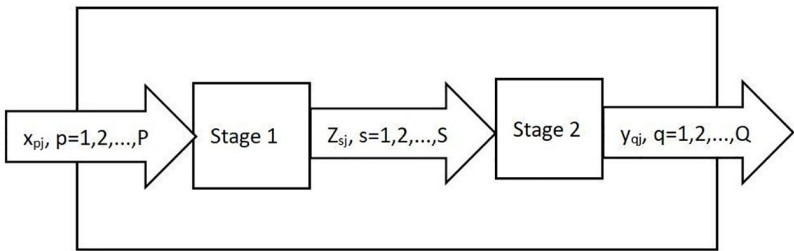

**Fig 1. Two-stage process of $DMU_j$.**

$X_j = (X_{1j}, X_{2j}, \ldots, X_{pj})$, $Y_j = (Y_{1j}, Y_{2j}, \ldots, y_q)$, $Z_j = (Z_{1j}, Z_{2j}, \ldots, Z_{sj})$, as the input vector, final output vector, and intermediate output vector for the $j$-th DMU, respectively. The weights for inputs, final outputs, and intermediate outputs are represented by $U = (u_1, u_2, \ldots, u_p)$, $V = (v_1, v_2, \ldots, v_q)$, $W = (w_1, w_2, \ldots, w_s)$ respectively. $OE_j$, $FE_j$, $SE_j$ represent the overall efficiency, first-stage efficiency, and second-stage efficiency of the $j$-th DMU, respectively [26, 27]. Thus, the calculation of two-stage DEA efficiency values is as follows:

$$OE_j = MAX\sum_{r=1}^{q} y_{rj}V_q / \sum_{i=1}^{p} x_{ij}U_p$$

$$FE_j = \sum_{m=1}^{s} z_{mj}W_s / \sum_{i=1}^{p} x_{ij}U_p \leq 1, j = 1, 2, \ldots, n$$

$$s.t. \quad SE_j = \sum_{r=1}^{q} y_{rj}V_q / \sum_{m=1}^{s} z_{mj}W_s \leq 1, j = 1, 2, \ldots, n$$

$$OE_j = \sum_{r=1}^{q} y_r V_q / \sum_{i=1}^{p} x_{ij}U_p \leq 1, j = 1, 2, \ldots, n$$

(1)

**3.1.2 Panel Tobit model.** After assessing the production efficiency of the real estate industry in different regions, this paper considers the efficiency values as dependent variables, and factors other than input-output data, reflecting the heterogeneity of the decision-making units as well as the regions, as independent variables. Drawing on insights from Amemiya, T. [28], Simar, L., and Wilson, P. W. [29], Banker, R. D., and Natarajan, R. regarding the analysis of influencing factors of restricted dependent variables, it is noted that "two-stage DEA-based procedures with OLS, ML, or even Tobit estimation in the second stage significantly outperform parametric methods" [30]. Given the practicality of computational software usage, this article employs a Tobit model to analyze the influencing factors affecting real estate production efficiency at this stage. The basic form of the model is presented in Eq (2). In this model, $OE_{it}$ represents the overall efficiency of the real estate industry in region i during year t. ALR and AD represent the asset-liability ratio and annual accumulated depreciation of firms, respectively, which collectively reflect the internal organizational and management conditions of the company; PP stands for permanent population, UR for urbanization rate, and PGDP for per capita income, and IS reflects the situation of industrial structure, all of which together demonstrate the socio-economic development environment of different regions; LIR represents the interest rate on housing provident fund loans with terms of five years or more, largely reflecting the orientation of real estate regulatory policies; PGA, PBA, and PRA respectively represent the per capita urban area of parks and green spaces, residential building area, and urban road area in different regions, which to some extent indicate the relative scarcity and allocation status of urban resources. $\beta_1 - \beta_{10}$ are the regression coefficients for these variables, $u_{it}$ is the

random error term. To minimize the impact of large differences in variable values on the analysis, natural logarithms are taken for variables with large values.

$$
\begin{aligned}
OE_{it} &= \beta_1 ALR_{it} + \beta_2 \ln AD_{it} + \beta_3 \ln PP_{it} + \beta_4 UR_{it} + \beta_5 \ln PGDP_{it} + \beta_6 IS_{it} \\
&\quad + \beta_7 LIR_{it} + \beta_8 PGA_{it} + \beta_9 PBA_{it} + \beta_{10} PRA_{it} + u_{it}
\end{aligned} \tag{2}
$$

**3.1.3 Threshold effects model.** Given the production efficiency in the first stage of the real estate sector tends to be stable across regions, whereas the second stage exhibits significant fluctuations, and considering the notable regional disparities in overall efficiency, this paper conducts a further analysis of the real estate market sales environment in the second stage across different regions. The per capita residential building area (PBA) is selected as the threshold variable for this analysis because it not only reflects the allocation of real estate resources across regions but also involves issues of appropriateness in scale. A panel threshold effects model is established to examine the impact and role of PBA on the production efficiency of the real estate industry. The basic expression of the model is shown in Eq (3).

$$
\begin{aligned}
OE_{it} &= \beta_1 \ln PP_{it} + \beta_2 UR_{it} + \beta_3 \ln PGDP_{it}(PBA \leq \gamma) \\
&\quad + \beta_4 \ln PGDP_{it}(PBA > \gamma) + \beta_5 IS_{it} + \beta_6 LIR_{it} \\
&\quad + \beta_7 PGA_{it} + \beta_8 PRA_{it} + u_{it}
\end{aligned} \tag{3}
$$

In Eq (3), PBA is the threshold variable, and $\gamma$ represents the threshold value, with the meanings of other variables consistent with those in model (2). The threshold effects model, established by Hansen in 1953 [31], analyzes situations where the intercept or slope of the model undergoes a sudden change at a certain time point by capturing the critical point or region of the threshold variable. This is achieved by considering each observation of the threshold variable as a potential threshold value and using the conditional least squares method to compute the sum of squared residuals and $S(\gamma)$ for these values. The model captures the observation that minimizes $S(\gamma)$, designating it as the threshold estimate $\hat{\gamma} = \arg\min \gamma S(\gamma)$; Subsequently, parameter estimates, the residual vector, and the corresponding sum of squared residuals are calculated. After obtaining parameter estimates, the significance of the threshold effects and threshold estimates is tested. Thus, model (3) can evaluate the dynamic heterogeneous effects on the overall efficiency of the real estate industry when PBA reaches the critical threshold value $\gamma$.

## 3.2 Variables and data

**3.2.1 Input variables.** The input variables for the real estate industry are based on the quantities of labor, land, and capital used. Therefore, the following indicators are selected for each region: the annual average number of employees in real estate development companies ($X_1$), land acquisition area ($X_2$), and total investment completed ($X_3$). Here, the average number of employees(people) reflects the labor input; land acquisition area (measured in ten thousand square meters) directly represents land input and usage; The total investment completed (measured in ten thousand CNY) encompasses the cumulative investment made by real estate development companies and other entities actively involved in real estate development or operations. This includes investments in various types of residential buildings, supporting service facilities, and land development projects, excluding pure land transaction activities.

**3.2.2 Intermediate variables.** Output and product value reflect the production outcomes at different stages of the real estate industry. Due to the long development and construction cycles in real estate and the difficulty in achieving market clearance in sales, the annual

completed housing area (Z, measured in ten thousand square meters) is designated as the intermediate variable. This helps to differentiate the resource utilization in various stages of the real estate industry.

**3.2.3 Output variables.** Focusing on measuring the utilization of real estate resources from the perspective of economic value, the output variables are set as the sales revenue from commercial housing ($Y_1$) and main business income ($Y_2$). Commercial housing sales revenue refers to the total contract price of commercial houses sold by real estate developers in each region annually, which is the price agreed upon in the official contracts signed by buyers and sellers, measured in hundred million CNY. Main business income includes revenues from commercial housing sales, land transfers, house rentals, etc., measured in ten thousand CNY.

**3.2.4 Dependent variables.** Using Dearun, the production efficiency of the real estate industry for the 31 provinces, municipalities, and autonomous regions of China from 2014–2022 is calculated. The DEA overall efficiency scores (OE) are used as the dependent variables when discussing the impact of various factors on production efficiency. For analyses focusing on phase-specific or regional differences, either the first stage efficiency scores (FE) or the second stage efficiency scores (SE) are used as dependent variables.

**3.2.5 Explanatory variables.** Based on the decomposition of real estate production efficiency by the two-stage DEA model, variables reflecting the internal organizational management conditions: asset-liability ratio (ALR, %) and annual accumulated depreciation (AD, measured in ten thousand CNY), and external market conditions: permanent population (PP), urbanization rate (UR, %), per capita GDP (PGDP, measured in CNY), and industrial structure (IS, represented by the proportion of GDP in the secondary and tertiary industries) are set as explanatory variables.

**3.2.6 Control variables.** To minimize the interference of individual effects on regression outcomes, variables reflecting the orientation of real estate regulatory policies and environmental conditions in different regions are used as control variables. These include the interest rate on personal housing provident fund loans with terms of five years or more (LIR, %) and per capita urban park and green space area (PGA, measured in square meters).

**3.2.7 Threshold variable.** Real estate sales conditions are influenced by the local market environments and, in particular, the configuration of existing housing stock. The PBA, not only provides a straightforward reflection of the regional housing stock configurations but also involves considerations of its appropriateness. Therefore, PBA (measured in square meters) is designated as a threshold variable to test the impact of regional heterogeneity on the production efficiency of the real estate industry. This approach helps to further determine the forms and extents of different factors affecting production efficiency in the real estate industry.

**3.2.8 Sample selection.** Given the gradual slowdown in the development pace of China's real estate industry and the fluctuating market environment faced by developers, data from 2014–2022 for the 31 provinces, municipalities, and autonomous regions are selected as the sample, totaling 279 observations. Descriptive statistics for each indicator are presented in Table 1. The primary data sources are the Wind database, with additional data on the real estate market and influencing factors derived from the "China Real Estate Yearbook" and "China Statistical Yearbook" (2015–2023). Production efficiency is calculated using Dearun.

# 4 Analysis of the production efficiency of the real estate industry in China's different regions

## 4.1 Overall efficiency

According to Table 2, the overall efficiency of the real estate industry across China's 31 provinces, municipalities, and autonomous regions is generally low, with Shanghai ranking first at

**Table 1. Descriptive statistics of relevant variables.**

| Variables | Names and symbols | Mean | Std. Dev. | Min | Max |
|---|---|---|---|---|---|
| Input Variables | Number of Employees (X1) | 89812.98 | 62005 | 1499 | 271392 |
| | Land Acquisition Area (X2) | 767.7 | 712.63 | 2.94 | 3776.54 |
| | Total Investment Completed (X3) | 3845.77 | 3274.99 | 40.37 | 17465.85 |
| Intermediate Variable | Completed Housing Area (Z) | 3165.20 | 2507.82 | 18.87 | 11373.68 |
| Output Variables | Commercial Housing Sales (Y1) | 4336.19 | 4401.47 | 21.08 | 22572.51 |
| | Commercial Housing Sales (Y2) | 3303.83 | 3446.87 | 20.23 | 17118.97 |
| Dependent Variables | Overall Efficiency (OE) | .153 | .091 | .046 | .494 |
| | First Stage Efficiency (FE) | .486 | .183 | .107 | 1 |
| | Second Stage Efficiency (SE) | .339 | .183 | .072 | 1 |
| Explanatory Variables | Asset-Liability Ratio (ALR) | 80.28 | 5.63 | 60.7 | 91.04 |
| | Annual Depreciation (AD) | 246759.9 | 222716.6 | 346.3 | 1202931.3 |
| | Permanent Population (PP) | 4510.61 | 2953 | 325 | 12684 |
| | Urbanization Rate (UR) | 61.09 | 12.02 | 26.23 | 89.33 |
| | Per Capita GDP (PGDP) | 65671.2 | 31153.8 | 26165.3 | 190313 |
| | Industrial Structure (IS) | 90.64 | 5.07 | 74.9 | 99.78 |
| Control Variables | Provident Fund Loan Rate (LIR) | 3.41 | .40 | 3.1 | 4.46 |
| | Per Capita Park Green Area (PGA) | 13.78 | 2.81 | 5.85 | 22.84 |
| Threshold Variables | Per Capita Residential Building Area (PBA) | 34.06 | 5.62 | 23.93 | 57.28 |
| | Per Capita Road Area (PRA) | 17.36 | 5.02 | 4.11 | 28 |

Data Sources: Wind Database, "China Real Estate Statistical Yearbook" and "China Statistical Yearbook" (2015–2023). The raw data is contained in S1 and S2 Data.

Software Used: Stata17

only 0.392. This indicates that the overall resource utilization in China's real estate industry from 2014 to 2022 is not optimistic, and improving production efficiency in this industry warrants deep consideration. Observing the annual averages, the production efficiency of China's real estate industry has gradually increased over these eight years, suggesting that regardless of the broader economic environment changes, the resource utilization efficiency within the real estate sector is continuously improving. When considering regional averages, there is a significant disparity in the production efficiency of the real estate industry among different areas. Regions like Shanghai, Zhejiang, Jiangsu, Beijing, Tianjin, Guangdong, and Hainan, which typically have higher levels of economic development, greater population densities, and stronger urban appeal, rank higher; whereas economically less developed regions with notable population outflows such as Xinjiang, Yunnan, Guizhou, Shanxi, Gansu, and Qinghai exhibit lower real estate production efficiencies. To further determine whether there is regional disparity in the overall efficiency of the real estate industry, the 31 provinces, municipalities, and autonomous regions are divided into Eastern, Central, and Western parts. As depicted in Fig 2, the production efficiency of the real estate industry in the Eastern region has consistently been higher than that in the Central and Western regions across all years. Although the Central and Western regions have a relatively small difference in their efficiencies, the Central region has always been slightly ahead of the Western region.

## 4.2 Phase efficiency

As indicated in Table 3, in most years and regions, the efficiency of the real estate industry during the first phase is higher than in the second phase, highlighting that from 2014 to 2022,

**Table 2. Overall efficiency of the real estate industry in China's 31 provinces, municipalities, and autonomous regions (Unit: %).**

| OE | 2014 | 2015 | 2016 | 2017 | 2018 | 2019 | 2020 | 2021 | 2022 | Average | Rank |
|---|---|---|---|---|---|---|---|---|---|---|---|
| Beijing | .157 | .170 | .223 | .202 | .228 | .302 | .280 | .409 | .449 | .269 | 4 |
| Tianjin | .162 | .175 | .242 | .275 | .233 | .227 | .238 | .328 | .355 | .248 | 5 |
| Hebei | .070 | .087 | .104 | .102 | .100 | .094 | .111 | .115 | .150 | .104 | 23 |
| Shanxi | .046 | .046 | .066 | .085 | .090 | .091 | .103 | .135 | .138 | .089 | 28 |
| Inner Mongolia | .064 | .070 | .089 | .083 | .114 | .095 | .111 | .125 | .162 | .102 | 25 |
| Liaoning | .094 | .102 | .117 | .144 | .169 | .179 | .219 | .206 | .219 | .161 | 8 |
| Jilin | .082 | .091 | .103 | .106 | .123 | .133 | .133 | .175 | .153 | .122 | 17 |
| Heilongjiang | .073 | .087 | .159 | .131 | .123 | .119 | .100 | .109 | .153 | .117 | 19 |
| Shanghai | .247 | .277 | .419 | .439 | .478 | .432 | .385 | .380 | .470 | .392 | 1 |
| Jiangsu | .155 | .174 | .249 | .257 | .347 | .312 | .356 | .385 | .414 | .294 | 3 |
| Zhejiang | .152 | .173 | .257 | .300 | .284 | .294 | .371 | .431 | .494 | .306 | 2 |
| Anhui | .086 | .094 | .129 | .128 | .155 | .159 | .177 | .205 | .217 | .150 | 10 |
| Fujian | .109 | .114 | .134 | .155 | .178 | .148 | .172 | .213 | .220 | .160 | 9 |
| Jiangxi | .102 | .099 | .126 | .131 | .148 | .148 | .159 | .199 | .220 | .148 | 11 |
| Shandong | .093 | .089 | .110 | .113 | .137 | .132 | .154 | .200 | .222 | .139 | 14 |
| Henan | .065 | .072 | .082 | .091 | .108 | .119 | .120 | .129 | .230 | .113 | 20 |
| Hubei | .073 | .089 | .114 | .132 | .172 | .156 | .136 | .194 | .223 | .143 | 13 |
| Hunan | .073 | .079 | .091 | .096 | .121 | .107 | .108 | .117 | .145 | .104 | 22 |
| Guangdong | .122 | .145 | .205 | .212 | .230 | .257 | .258 | .281 | .283 | .221 | 6 |
| Guangxi | .059 | .073 | .075 | .089 | .111 | .111 | .106 | .121 | .180 | .103 | 24 |
| Hainan | .061 | .061 | .094 | .181 | .163 | .304 | .248 | .179 | .247 | .171 | 7 |
| Chongqing | .095 | .092 | .097 | .111 | .148 | .150 | .155 | .164 | .153 | .129 | 15 |
| Sichuan | .088 | .088 | .102 | .128 | .143 | .149 | .164 | .186 | .261 | .146 | 12 |
| Guizhou | .046 | .055 | .078 | .087 | .087 | .087 | .092 | .098 | .158 | .088 | 29 |
| Yunnan | .056 | .054 | .062 | .071 | .092 | .089 | .090 | .096 | .154 | .085 | 30 |
| Tibet | .049 | .047 | .103 | .141 | .105 | .107 | .078 | .152 | .304 | .121 | 18 |
| Shaanxi | .073 | .071 | .094 | .094 | .095 | .114 | .133 | .159 | .273 | .123 | 16 |
| Gansu | .063 | .074 | .097 | .114 | .092 | .092 | .088 | .107 | .130 | .095 | 27 |
| Qinghai | .057 | .056 | .164 | .075 | .078 | .096 | .102 | .159 | .105 | .099 | 26 |
| Ningxia | .070 | .062 | .082 | .079 | .091 | .111 | .129 | .170 | .164 | .107 | 21 |
| Xinjiang | .076 | .072 | .097 | .086 | .077 | .078 | .079 | .092 | .075 | .081 | 31 |
| Average | .091 | .098 | .134 | .143 | .155 | .161 | .166 | .194 | .230 | | |

Data Sources: Wind Database, "China Real Estate Statistical Yearbook" and "China Statistical Yearbook" (2015–2023). Please refer to S1 Date for the raw input-output data of China's real estate industry. Software Used: Dearun

China has managed its resources better during the real estate development stage. However, due to factors such as economic instability and lengthy real estate de-stocking cycles, the efficiency of the second phase is generally lower. Consequently, whether there is overcapacity in China's real estate industry and how to enhance the production efficiency of the second phase are questions that merit thorough exploration. Looking at the annual averages, the efficiencies of the first and second phases have fluctuated significantly across different years. While the efficiency of the first phase is often higher than that of the second phase, a reversal occurred in 2020. This change was likely due to the impact of COVID-19 prevention policies that year, which significantly restricted real estate development. In contrast, real estate sales were less affected by these controls and did not rely on intermediate products' rapid growth, leading to an improvement in the second phase's efficiency. Analysis reveals that within the two-stage

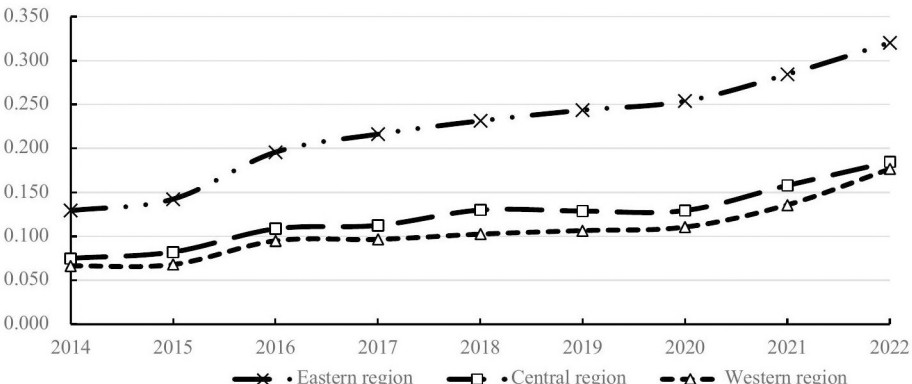

**Fig 2. Overall efficiency of the real estate industry in the Eastern, Central, and Western Regions (Unit: %).** Data Sources: Wind Database, "China Real Estate Statistical Yearbook" and "China Statistical Yearbook" (2015–2023) Software Used: Dearun.

DEA model, the relationship between the efficiency values is expressed as $OE_j = FE_j \times SE_j$. This formula indicates that the overall efficiency of the real estate industry is determined by the combined efficiencies of different stages. Therefore, to enhance the overall production efficiency in various regions, it is essential to consider not only the resource utilization during the development and construction phase but also to pay attention to the socio-economic conditions of the region during the sales phase, the impacts of macroeconomic regulatory policies, and most crucially, to effectively manage the flow of resources from stages of development through to sales.

# 5 Analysis of factors affecting production efficiency in China's real estate industry

## 5.1 Baseline regression results from the Tobit model

When using Model (2) to assess the impact of various factors on real estate production efficiency, we discovered that internal company factors: the asset-liability ratio show a significantly negative correlation with overall efficiency, while annual accumulated depreciation exhibits a positive correlation with overall efficiency. This indicates that a business development model characterized by "lighten up and make a bold move" is beneficial for enhancing the production efficiency of the real estate industry see Table 4. External Factors: The impact of a city's permanent population on production efficiency is not significant; however, increased urbanization levels can enhance overall efficiency but diminish efficiency in the second phase, indicating that higher urbanization levels may hinder real estate sales but can boost real estate production efficiency in the long term. The growth in per capita GDP significantly enhances real estate production efficiency, whereas the influence of industrial structure is weak and not significant. Additionally, the provident fund loan rate has a negative correlation with real estate production efficiency, suggesting that stringent real estate regulatory policies are detrimental to improving production efficiency in the industry. The per capita area of park green spaces and urban road spaces are negatively correlated with real estate production efficiency, indicating that improvements in urban environment and public resource allocation come at the cost of reduced real estate production efficiency. Conversely, an increase in per capita residential building area significantly boosts real estate production efficiency,

**Table 3. The two-stage efficiency of China's real estate industry from 2014 to 2022 (Unit: %).**

| Regions / Stages | 2014 FE | 2014 SE | 2015 FE | 2015 SE | 2016 FE | 2016 SE | 2017 FE | 2017 SE | 2018 FE | 2018 SE | 2019 FE | 2019 SE | 2020 FE | 2020 SE | 2021 FE | 2021 SE | 2022 FE | 2022 SE |
|---|---|---|---|---|---|---|---|---|---|---|---|---|---|---|---|---|---|---|
| Beijing | .42 | .38 | .38 | .45 | .37 | .60 | .24 | .86 | .33 | .69 | .38 | .79 | .44 | .64 | .56 | .73 | .64 | .70 |
| Tianjin | 1.0 | .16 | 1.0 | .18 | 1.0 | .24 | .73 | .38 | .75 | .31 | .57 | .40 | .61 | .39 | .75 | .44 | .76 | .47 |
| Hebei | .44 | .16 | .47 | .18 | .48 | .22 | .39 | .27 | .26 | .38 | .30 | .31 | .27 | .41 | .29 | .39 | .45 | .33 |
| Shanxi | .53 | .09 | .49 | .09 | .66 | .10 | .59 | .14 | .39 | .23 | .64 | .14 | .33 | .31 | .62 | .22 | .57 | .24 |
| Inner Mongolia | .53 | .12 | .54 | .13 | .58 | .16 | .65 | .13 | .56 | .20 | .36 | .27 | .31 | .36 | .41 | .31 | .57 | .28 |
| Liaoning | .77 | .12 | .50 | .21 | .49 | .24 | .57 | .25 | .49 | .34 | .42 | .43 | .45 | .48 | .59 | .35 | .59 | .37 |
| Jilin | .52 | .16 | .47 | .19 | .47 | .22 | .55 | .19 | .56 | .22 | .49 | .27 | .43 | .31 | .42 | .42 | .40 | .38 |
| Heilongjiang | .78 | .09 | 1.0 | .09 | 1.0 | .16 | .69 | .19 | .44 | .28 | .46 | .26 | .55 | .18 | .43 | .26 | .47 | .32 |
| Shanghai | .44 | .57 | .54 | .52 | .53 | .79 | .78 | .56 | .86 | .56 | .74 | .59 | .63 | .61 | .61 | .62 | .47 | 1.0 |
| Jiangsu | .63 | .25 | .70 | .25 | .71 | .35 | .68 | .38 | .60 | .58 | .67 | .47 | .77 | .46 | .64 | .60 | .69 | .60 |
| Zhejiang | .62 | .25 | .60 | .29 | .84 | .31 | .72 | .42 | .51 | .56 | .54 | .55 | .63 | .59 | .64 | .68 | .72 | .69 |
| Anhui | .56 | .16 | .62 | .15 | .63 | .21 | .51 | .25 | .45 | .35 | .55 | .29 | .52 | .34 | .75 | .27 | .78 | .28 |
| Fujian | .46 | .24 | .44 | .26 | .48 | .28 | .55 | .28 | .45 | .40 | .31 | .47 | .45 | .38 | .53 | .40 | .60 | .37 |
| Jiangxi | .48 | .21 | .43 | .23 | .33 | .39 | .32 | .41 | .34 | .44 | .35 | .42 | .35 | .46 | .42 | .47 | .32 | .70 |
| Shandong | .48 | .19 | .51 | .18 | .47 | .23 | .47 | .24 | .58 | .24 | .59 | .23 | .55 | .28 | .69 | .29 | .48 | .46 |
| Henan | .60 | .11 | .44 | .17 | .43 | .19 | .40 | .23 | .42 | .26 | .42 | .29 | .34 | .35 | .49 | .27 | .99 | .23 |
| Hubei | .32 | .23 | .29 | .31 | .33 | .35 | .33 | .40 | .26 | .67 | .24 | .64 | .27 | .50 | .34 | .56 | .42 | .53 |
| Hunan | .49 | .15 | .52 | .15 | .53 | .17 | .42 | .23 | .38 | .32 | .33 | .33 | .35 | .31 | .41 | .29 | .47 | .31 |
| Guangdong | .40 | .30 | .34 | .43 | .36 | .58 | .41 | .52 | .36 | .64 | .48 | .54 | .38 | .69 | .40 | .70 | .48 | .59 |
| Guangxi | .35 | .17 | .33 | .22 | .28 | .27 | .28 | .32 | .32 | .35 | .26 | .42 | .27 | .39 | .33 | .36 | .51 | .36 |
| Hainan | .36 | .17 | .32 | .19 | .49 | .19 | .41 | .44 | .40 | .40 | 1.0 | .30 | .52 | .48 | .20 | .90 | .48 | .51 |
| Chongqing | .46 | .21 | .59 | .16 | .55 | .18 | .59 | .19 | .50 | .30 | .64 | .23 | .46 | .34 | .54 | .30 | .49 | .32 |
| Sichuan | .48 | .19 | .41 | .22 | .63 | .16 | .52 | .25 | .44 | .32 | .36 | .41 | .36 | .46 | .36 | .53 | .55 | .47 |
| Guizhou | .49 | .09 | .44 | .13 | .38 | .20 | .20 | .43 | .19 | .46 | .13 | .66 | .13 | .73 | .14 | .72 | .30 | .52 |
| Yunnan | .27 | .21 | .39 | .14 | .35 | .18 | .34 | .21 | .20 | .46 | .22 | .41 | .19 | .48 | .32 | .30 | .51 | .30 |
| Tibet | .34 | .14 | .66 | .07 | .43 | .24 | .38 | .37 | .35 | .30 | .11 | 1.0 | .13 | .59 | .37 | .41 | .30 | 1.00 |
| Shaanxi | .37 | .20 | .28 | .25 | .42 | .23 | .37 | .25 | .22 | .44 | .26 | .43 | .27 | .50 | .31 | .51 | .66 | .41 |
| Gansu | .38 | .17 | .43 | .17 | .47 | .21 | .42 | .27 | .28 | .32 | .26 | .36 | .29 | .30 | .50 | .21 | .41 | .32 |
| Qinghai | .66 | .09 | .62 | .09 | .79 | .21 | .57 | .13 | .44 | .18 | .18 | .55 | .21 | .49 | .23 | .69 | .38 | .28 |
| Ningxia | .78 | .09 | .80 | .08 | .93 | .09 | 1.0 | .08 | 1.0 | .09 | .89 | .13 | .65 | .20 | 1.0 | .17 | .61 | .27 |
| Xinjiang | .70 | .11 | .55 | .13 | .62 | .16 | .56 | .15 | .41 | .19 | .38 | .21 | .27 | .30 | .42 | .22 | .35 | .21 |
| Average | .52 | .19 | .52 | .20 | .55 | .26 | .50 | .30 | .44 | .37 | .44 | .41 | .40 | .43 | .47 | .44 | .53 | .45 |

Data Sources: Wind Database, "China Real Estate Statistical Yearbook" and "China Statistical Yearbook" (2015–2023). Please refer to S1 Dearun Date for the raw input-output data of China's real estate industry.Software Used: Dearun

underscoring that demand for improved living conditions is a potent driver for the sustained development of the real estate industry. In order to determine what factors affect the production efficiency of the first and second stages of the real estate industry, this article once again sets FE as the dependent variable and ALR and AD as the explanatory variables; SE as the dependent variable, while PP, UR, PGDP, and IS are the explanatory variables. The simplified model (2) is used to discuss the effects of the internal organization and management of real estate development enterprises and the external market and sales environment on the production efficiency at different stages. Although the regression results are somewhat different from model (2), however, the role of the main explanatory variables on OE, FE and SE was further verified.

**Table 4. The impact of various factors on production efficiency in the real estate industry.**

|  | OE | FE | SE |
|---|---|---|---|
| ALR | -0.00224*** | -0.00632** |  |
|  | (-4.20) | (-2.95) |  |
| lnAD | 0.0147* | 0.00338 |  |
|  | (2.49) | (0.30) |  |
| lnPP | -0.0104 |  | -0.00197 |
|  | (-1.75) |  | (-0.20) |
| UR | 0.00112* |  | -0.00507*** |
|  | (2.12) |  | (-3.43) |
| lnPGDP | 0.0955*** |  | 0.355*** |
|  | (5.87) |  | (7.31) |
| IS | 0.000440 |  | -0.00539* |
|  | (0.55) |  | (-2.27) |
| LIR | -0.0397*** | 0.043 | -0.121*** |
|  | (-5.51) | (1.51) | (-5.46) |
| PGA | -0.00577*** | 0.00359 | -0.00720* |
|  | (-5.00) | (0.75) | (-2.43) |
| PBA | 0.00372*** | 0.00459* | 0.00565*** |
|  | (6.44) | (2.09) | (3.37) |
| PRA | -0.0000914 | 0.0023 | -0.00999*** |
|  | (-0.13) | (0.86) | (-4.54) |
| _cons | -0.829*** | 0.562* | -2.341*** |
|  | (-5.52) | (2.27) | (-5.72) |
| Var(e) | 0.00180*** | 0.0325*** | 0.0179*** |
|  | (11.81) | (11.52) | (11.70) |
| Pseudo r2 | 0.771 | 0.176 | 0.861 |
| N | 279 | 279 | 279 |

Note:

***, **, and * denote statistical significance at the 1%, 5%, and 10% levels, respectively, with t-test values provided in brackets. Please refer to S2 Date for the raw data of Productivity and influencing Factors in China's Real Estate Industry.

## 5.2 Threshold effect test and regression results

To explore the impact of regional heterogeneity on the production efficiency of the real estate industry, this study employs Model (3) to examine whether per capita residential building area exhibits a threshold effect on production efficiency. According to the Bootstrap method test results of panel threshold fixed effects (Table 5), the per capita residential building area shows a significant single threshold effect and passes the significance test at the 5% level. Based on

**Table 5. Threshold effect test (bootstrap = 300 300 300).**

| Threshold | Fstat | Prob | Crit10 | Crit5 | Crit1 |
|---|---|---|---|---|---|
| Single | 30.98** | 0.01 | 20.983 | 24.599 | 28.877 |
| Double | 19.92* | 0.08 | 18.535 | 21.584 | 34.706 |
| Triple | 12.79 | 0.28 | 17.487 | 22.768 | 33.487 |

Note:

** and * indicate significance at the 5% and 10% levels, respectively, with P-values and F-values obtained after 300 Bootstrap method.

**Table 6. Threshold regression results.**

|  | OE |
| --- | --- |
| lnPP | -0.0626 |
|  | (-0.49) |
| UR | -0.00103 |
|  | (-0.61) |
| lnPGDP ($PBA \leq 45.62$) | 0.187*** |
|  | (7.54) |
| lnPGDP ($PBA > 45.62$) | 0.602*** |
|  | (7.30) |
| IS | -0.00176 |
|  | (-0.67) |
| LIR | -0.0304*** |
|  | (-3.82) |
| PGA | -0.0016 |
|  | (-0.58) |
| PRA | -0.00133 |
|  | (-0.88) |
| _cons | -1.178 |
|  | (-1.28) |
| $R^2$ | 0.668 |
| N | 279 |

Note:

***, **, and * denote statistical significance at the 1%, 5%, and 10% levels, respectively, with t-test values provided in brackets.

300 Bootstrap tests, the threshold estimate is 45.62 square meters, with a 95% confidence interval of [31.22, 48.48]. This finding indicates that the configuration of existing housing stock has a dynamically heterogeneous effect on real estate production efficiency and underscores that regional characteristics of the real estate market are one of the causes for variations in production efficiency.

From the panel threshold effect parameter estimates (Table 6), the main explanatory variables and control variables essentially retain their influence on real estate production efficiency, though the impact levels vary slightly. The threshold variable PBA has a significant single threshold effect on production efficiency. When PBA ≤ 45.62, a 1% increase in per capita GDP leads to an 18.7% increase in overall real estate efficiency. When PBA >45.62, a 1% increase in per capita GDP results in a 60.2% increase in overall efficiency. This suggests that the impact of regional economic development on enhancing real estate production efficiency changes dramatically when the per capita residential building area reaches 45.62 square meters. As demand for existing housing stock increases, real estate production efficiency significantly improves. This can also be interpreted as improved living conditions driven by higher income levels being a primary force for the sustained development of the real estate industry.

### 5.3 Robustness test

To avoid spurious regression, this study further tests whether the per capita residential building area also exhibits a dynamically heterogeneous effect on another explanatory variable-the

**Table 7. Robustness test results.**

| | OE |
|---|---|
| lnPP<br>($PBA \leq 45.62$) | -0.0819 |
| | (-0.64) |
| lnPP<br>(PBA > 45.62) | 1.778*** |
| | (5.58) |
| UR | -0.0012 |
| | (-0.71) |
| lnPGDP | 0.189*** |
| | (7.66) |
| IS | -0.00164 |
| | (-0.63) |
| LIR | -0.0309*** |
| | (-3.91) |
| PGA | -0.00148 |
| | (-0.54) |
| PRA | -0.0013 |
| | (-0.87) |
| _cons | -1.409 |
| | (-1.56) |
| $R^2$ | 0.652 |
| N | 279 |

Note:

***, **, and * denote statistical significance at the 1%, 5%, and 10% levels, respectively, with t-test values provided in brackets.

permanent population. The robustness test results (Table 7) indicate a significant reversal in the impact of population growth on real estate production efficiency when PBA reaches 45.62 square meters. Specifically, when PBA $\leq$ 45.62, a 1% increase in permanent population leads to an 8.19% decrease in overall real estate efficiency. Conversely, when PBA >45.62, a 1% increase in permanent population causes a 177.8% increase in overall efficiency. That is, when the per capita residential building area reaches 45.62 square meters, the negative effect of the increase of regional permanent population on the production efficiency of the real estate industry suddenly changes into a positive effect. Under the comprehensive effect of the increase of population and the increase of per capita residential building area, the production efficiency of the real estate industry can be greatly improved. It can also be understood that more residents and greater per capita housing demand are the main way to improve the production efficiency of the real estate industry. During the panel threshold effect analysis and robustness testing, we identified a significant single threshold effect of per capita residential building area on real estate production efficiency. This finding not only further elucidates the reasons behind the variations in real estate production efficiency across different times and regions, but also offers a new avenue for addressing the generally low production efficiency prevalent in China's real estate industry today.

## 6 Conclusions and implications

The analysis of real estate production efficiency from 2014 to 2022 across China's 31 provinces, municipalities, and autonomous regions provides an objective reflection of the utilization of

its real estate resources. Based on the characteristics of efficiency across different phases and regions, it is evident that internal organization and management of enterprises and the external market environment for real estate sales are the primary drivers of these variations. Influential factors include the asset-liability ratios and annual accumulated depreciation of real estate development companies, permanent population, per capita GDP, real estate regulatory policies, and public resource allocation, all reflecting the socio-economic development status of the regions. The efficiency values of the first and second stages indicate that improving the real estate sales environment, enhancing market liquidity, stimulating demand for improved living conditions, and accelerating property turnover are key ways to boost production efficiency in the real estate industry. The depth and novelty of this study could be further enhanced.

Based on these findings, the following recommendations are proposed:

Firstly, in regions where overall real estate efficiency is low, regional attributes of the real estate market should be fully considered. Objective and accurate evaluation of resource utilization in real estate development and sales phases is crucial. Identifying existing issues promptly will prepare the groundwork for the sustainable and healthy development of the real estate industry in various regions.

Secondly, the regression coefficients relating to the asset-liability ratio and annual accumulated depreciation of real estate development companies indicate that lower debt levels and higher depreciation are beneficial for enhancing the production efficiency of the real estate industry. Therefore, it is advisable for real estate development companies to adopt a development model characterized by "lighten up and make a bold move".

Thirdly, the production efficiency of the real estate industry is significantly influenced by the socio-economic development status of regions. When considering how to improve the production efficiency of the real estate industry, in addition to paying attention to the input-output data of the real estate industry itself, it is also necessary to think about economic growth and regional development issues under the macro framework.

Fourthly, focus on improving efficiency in the sales phase: inadequate market liquidity has been a fundamental reason for the generally low production efficiency of China's real estate industry in recent years. Administrative bodies could implement preferential policies, reduce transaction costs appropriately, and stimulate demand for high-quality new housing, thereby accelerating property replacement and improving production efficiency of the real estate industry.

Fifthly, supply without demand constitutes a waste of resources. The two-stage DEA model can accurately evaluate the resource utilization status from production to sales in the real estate industry. Developers should consider establishing a "demand-supply linkage" model, scientifically adjusting the number of real estate developments based on market demand. Furthermore, increasing the per capita residential building area and leveraging the demand for improved living conditions serve as powerful drivers to enhance the production efficiency of the real estate industry. This approach is crucial for ensuring the sustained and healthy development of the real estate sector.

## Supporting information

**S1 Data. Dearun data.**
(XLSX)

**S2 Data. Tobit data.**
(XLSX)

## Author Contributions

**Conceptualization:** Jiening Meng.

**Data curation:** Jiening Meng.

**Formal analysis:** Jiening Meng, Wei Bu.

**Investigation:** Jiening Meng.

**Methodology:** Jiening Meng, Wei Bu.

**Project administration:** Wei Bu.

**Resources:** Wei Bu.

**Supervision:** Wei Bu.

**Writing – original draft:** Jiening Meng.

**Writing – review & editing:** Wei Bu.

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
