## [Decision Letter · Decision Letter 0]

5 Mar 2024

PONE-D-23-38796Analysis of Total Factor Productivity in the Real Estate Industry of 35 Cities in China Based on the Malmquist IndexPLOS ONE

Dear Dr. Meng,

Thank you for submitting your manuscript to PLOS ONE. After careful consideration, we feel that it has merit but does not fully meet PLOS ONE’s publication criteria as it currently stands. Therefore, we invite you to submit a revised version of the manuscript that addresses the points raised during the review process.

We look forward to receiving your revised manuscript.

Kind regards,

Ahmad Hassan Ahmad

Academic Editor

PLOS ONE

Journal Requirements:

Reviewers' comments:

Reviewer's Responses to Questions

**Comments to the Author**

1. Is the manuscript technically sound, and do the data support the conclusions?

Reviewer #1: No

Reviewer #2: Partly

2. Has the statistical analysis been performed appropriately and rigorously? 

Reviewer #1: N/A

Reviewer #2: Yes

3. Have the authors made all data underlying the findings in their manuscript fully available?

Reviewer #1: Yes

Reviewer #2: Yes

4. Is the manuscript presented in an intelligible fashion and written in standard English?

Reviewer #1: No

Reviewer #2: Yes

5. Review Comments to the Author

Reviewer #1: This study analyzed the total factor productivity in the real estate industry of 35 Cities in China using the Malmquist index.

Generally, I think this paper did not reach the publication level due the theoretical contribution nor the academic writing. My comments are as follows:

1. The literature review should be focused on the similar study in how to analyze the total factor productivity in different countries, regions, industries. However, no work on this area.

2. Regarding the methods, I think the authors should focus on the why the DEA Malmquist index model suits this study, rather than show what is the DEA Malmquist index model by 3-4 pages.

3. Data source is the base of this study. However, how to obtain the relevant data is not clarified.

Reviewer #2: 1. The DEA Malmquist index model measures production efficiency. The selection of input factors is very important. How to select and decide?

2. The real estate industry is greatly affected by overall economic and policy factors. How does this model deal with these two factors?

3. Generally speaking, the larger the R square, the better the fit between the model and the data. The R square in Table 5 is only between 10.8% and 12.9%, indicating that the model fit in this article is low and needs to be re-examined.

4. What impact does the result of this article have on the existing "purchase restriction orders" in first-tier cities and the "destocking" policies in third- and fourth-tier cities?

6. PLOS authors have the option to publish the peer review history of their article (what does this mean?). If published, this will include your full peer review and any attached files.

Reviewer #1: No

Reviewer #2: No

---

## [Author Response · Author response to Decision Letter 0]

3 May 2024

Response to Reviewers

Dear reviewer:

I hope all is well when you read the letter!

Thank you very much for taking the time to review this manuscript. We have made the following reflections or explanations for your questions. 

Please find the detailed responses below and the corresponding revisions in the re-submitted files.

 Jiening Meng

 From the ancient capital of Luoyang

April 21, 2024

Review Comments to the Author

Reviewer #1: 

1. The literature review should be focused on the similar study in how to analyze the total factor productivity in different countries, regions, industries. However, no work on this area.

Indeed, there are such issues in the original paper, but when we use the DEA Malmquist index model to decompose the total factor productivity (TFP) of the real estate industry into: pure technical efficiency, technological progress efficiency, and scale efficiency, we find that the improvements in technological progress efficiency are the main driver of TFP growth in the real estate industry across 35 major and medium-sized cities in China. We believe that this offers little significance in addressing current issues such as the lack of liquidity and severe regional disparities in China's real estate market. What we should consider now is: Why is there such a large disparity in the production efficiency of the real estate industry across different regions in China? What factors are limiting the improvement of production efficiency in China's real estate industry?

Considering the recent real estate investment and sales conditions in different regions of China, this paper has opted to utilize a two-stage DEA model to measure the production efficiency of the real estate industry in 31 provinces, municipalities, and autonomous regions of China (due to data availability constraints), and to evaluate their resource utilization during the development and sales stages. The results were enlightening. 

Additionally, in the updated literature review, we have shifted away from merely listing literature by Chinese scholars or studies related to the production efficiency of the Chinese real estate industry. Instead, we have structured it around the thematic thread of "distinct issues in real estate markets - spatial discrepancies in real estate prices - regional variations in real estate production efficiency," summarizing relevant achievements and shortcomings. Furthermore, in the section discussing models and methods, a brief overview of relevant issues concerning efficiency estimation is provided when discussing why a two-stage DEA model was chosen and features of its analysis method.

2. Regarding the methods, I think the authors should focus on the why the DEA Malmquist index model suits this study, rather than show what is the DEA Malmquist index model by 3-4 pages.

After 2014, the most significant challenge facing the development of China's real estate industry has been the phenomenon of regional differentiation in real estate markets. To better understand the utilization of real estate resources in different regions and reveal the reasons for their disparities, this paper has switched to employing a two-stage DEA model to measure the production efficiency of the real estate industry in various regions and to thoroughly assess their resource utilization during the development and sales stages. The analysis results indicate that the low efficiency and significant gap during the sales stage are the primary reasons for the regional differences in the production efficiency of the Chinese real estate industry, which holds crucial significance for effectively addressing related issues.

Given the adoption of a new method, the two-stage DEA model, all explanations regarding the DEA-Malmquist index model from the original paper have been omitted.

3.Data source is the base of this study. However, how to obtain the relevant data is not clarified.

We have added data sources in the revised draft:

The primary data sources are the Wind database, with additional data on the real estate market and influencing factors derived from the "China Real Estate Yearbook" and "China Statistical Yearbook" (2015-2023). Production efficiency is calculated using Dearun.

Reviewer #2: 

1.The DEA Malmquist index model measures production efficiency. The selection of input factors is very important. How to select and decide?

Neoclassical economics categorizes all production factors into four types: land, capital, labor, and entrepreneurial talent. Following this line and considering the availability of relevant statistical data in China, this paper focuses on analyzing the quantities of people, land, and money utilized in the processes of real estate development and sales. 

2.The real estate industry is greatly affected by overall economic and policy factors. How does this model deal with these two factors?

In the process of calculating production efficiency, we only consider the quantities of input factors and output products, as DEA generally treats the production process as a "black box." However, when analyzing the factors affecting the production efficiency of the real estate industry using the Tobit model and threshold effect model, we incorporate relevant indicators reflecting macroeconomic conditions, regional economic and social development, orientations of real estate regulation policies, and internal organizational management status of real estate development enterprises as explanatory or control variables. The regression coefficients of these variables indicate significant effects among them.

3.Generally speaking, the larger the R square, the better the fit between the model and the data. The R square in Table 5 is only between 10.8% and 12.9%, indicating that the model fit in this article is low and needs to be re-examined.

Indeed, the original paper encounters the issues you mentioned when analyzing the roles of various influencing factors using the Moderating effect model. After careful consideration, we found that efficiency values are constants between 0 and 1. Hence, when analyzing their influencing factors, a constrained dependent variable model, the Tobit model, should be utilized. However, due to the limited availability of data concerning the real estate industry in 35 major cities in China, the direct application of the Tobit model for data analysis yielded unsatisfactory results. Consequently, this paper had to substitute samples and data, gathering relevant data on the real estate industry in 31 provinces, municipalities, and autonomous regions of China from 2014 to 2022. The utilization of the Tobit model for regression improved significantly, achieving a fit of around 70%, compared to previous attempts.

4.What impact does the result of this article have on the existing "purchase restriction orders" in first-tier cities and the "destocking" policies in third- and fourth-tier cities?

The "purchase restriction orders" and "destocking" measures are primarily mentioned to illustrate that without understanding the underlying reasons for the lower production efficiency of the real estate industry in various regions, government departments tend to focus solely on formulating policies related to real estate sales. However, this approach not only fails to fundamentally alter the phenomenon of irrational allocation of real estate resources but also constitutes excessive government intervention in the operation of the real estate market. If the unreasonable utilization of resources in the real estate development or sales stage can be identified promptly, adjustments in the quantities of input factors at each stage can be made in a timely manner. By reducing the number of real estate developments and minimizing the creation of inventory, the issue of misallocation of real estate resources can be addressed at its root.

---

## [Decision Letter · Decision Letter 1]

6 Aug 2024

PONE-D-23-38796R1An   Analysis of Production Efficiency in China's Real Estate Industry Based on a Two-Stage DEA ModelPLOS ONE

Dear Dr. Meng,

Thank you for submitting your manuscript to PLOS ONE. After careful consideration, we feel that it has merit but does not fully meet PLOS ONE’s publication criteria as it currently stands. Therefore, we invite you to submit a revised version of the manuscript that addresses the points raised during the review process.

Please submit your revised manuscript by Sep 20 2024 11:59PM If you will need more time than this to complete your revisions, please reply to this message or contact the journal office at plosone@plos.org. Please include the following items when submitting your revised manuscript:A rebuttal letter that responds to each point raised by the academic editor and reviewer(s). You should upload this letter as a separate file labeled 'Response to Reviewers'.A marked-up copy of your manuscript that highlights changes made to the original version. You should upload this as a separate file labeled 'Revised Manuscript with Track Changes'.An unmarked version of your revised paper without tracked changes. You should upload this as a separate file labeled 'Manuscript'.If applicable, we recommend that you deposit your laboratory protocols in protocols.io to enhance the reproducibility of your results. Protocols.io assigns your protocol its own identifier (DOI) so that it can be cited independently in the future. For instructions see: https://journals.plos.org/plosone/s/submission-guidelines#loc-laboratory-protocols. Additionally, PLOS ONE offers an option for publishing peer-reviewed Lab Protocol articles, which describe protocols hosted on protocols.io. Read more information on sharing protocols at https://plos.org/protocols?utm_medium=editorial-email&utm_source=authorletters&utm_campaign=protocols.

We look forward to receiving your revised manuscript.

Kind regards,

Ahmad Hassan Ahmad

Academic Editor

PLOS ONE

Journal Requirements:

Additional Editor Comments:

Based on the comments, the paper is given minor revisions. This is an opportunity for you to revise the paper addressing all the comments raised carefully and satisfactorily.

Reviewers' comments:

Reviewer's Responses to Questions

**Comments to the Author**

1. If the authors have adequately addressed your comments raised in a previous round of review and you feel that this manuscript is now acceptable for publication, you may indicate that here to bypass the “Comments to the Author” section, enter your conflict of interest statement in the “Confidential to Editor” section, and submit your "Accept" recommendation.

Reviewer #3: (No Response)

2. Is the manuscript technically sound, and do the data support the conclusions?

Reviewer #3: Yes

3. Has the statistical analysis been performed appropriately and rigorously? 

Reviewer #3: Yes

4. Have the authors made all data underlying the findings in their manuscript fully available?

Reviewer #3: Yes

5. Is the manuscript presented in an intelligible fashion and written in standard English?

Reviewer #3: Yes

6. Review Comments to the Author

Reviewer #3: This paper uses a two-stage Data Envelopment Analysis (DEA) model to calculate the production efficiency of China's real estate industry across 31 regions in 2014-2022. It uses a panel Tobit model to identify factors (the leverage ratio of companies, per capita GDP and regulatory policies) that significantly impact efficiency.

In Eq. 2, the methodology uses Tobit regression to explain efficiency. There are other regression methods such as truncated regression, robust-OLS regression or Papke-Wooldridge which are probably more suited method for these purposes. Adding a short discussion of these alternative ways for this type of analysis will be helpful (see Simar and Wilson 2008, Banker and Natarajan 2007 for discussion).

Banker R.D and Natarajan R. (2008) Evaluating Contextual Variables Affecting Productivity Using Data Envelopment Analysis, Operations Research, Vol. 56, No. 1, January–February 2008, pp. 48–58

Papke L.E. and Wooldridge J.M., (1996) Econometric methods for fractional response variables with an application to 401(k) plan participation rates, Journal of Applied Econometrics, 11 (6) (1996), pp. 619–632

Simar L, Wilson PW. (2008) Statistical inference in non-parametric frontier models: recent developments and perspectives. In Fried HO, Knox-Lovell CA, Schmidt S, The measurement of productive efficiency and productivity growth. 2008; Oxford University Press, Oxford.

7. PLOS authors have the option to publish the peer review history of their article (what does this mean?). If published, this will include your full peer review and any attached files.

Reviewer #3: No

---

## [Author Response · Author response to Decision Letter 1]

12 Aug 2024

Dear editor and reviewer,

I am writing on behalf of all the contributing author to express our sincere appreciation for your letter and the constructive comments provided by the reviewer regarding our article entitled "An Analysis of Production Efficiency in China's Real Estate Industry Based on a Two-Stage DEA Model" (Manuscript No.: PONE-D-23-38796R1). These comments are extremely valuable and have greatly contributed to improving our article. In response to these comments, extensive modifications have been made to the manuscript. In this revised version, all changes have been highlighted in red within the document. A point-by-point response to the editor and reviewer is provided below this letter.

Thank you very much for your time and consideration.

Best regards,

Jiening Meng

Corresponding author:

Name: Jiening Meng

E-mail: 16113105@bjtu.edu.cn

Responds to Journal Requirements:

We have thoroughly examined all the literature referenced in the article, and fortunately, there are no retracted articles. The citation and organization of the literature are also arranged in accordance with the journal's requirements. However, when we modified the content of section 3.1.2 based on the reviewer's suggestion, we replaced the 28th and 29th references and added a new one at the same time, resulting in an additional reference in the revised draft compared to the original.

Responds to Additional Editor Comments:

Based on the comments, the paper is given minor revisions. This is an opportunity for you to revise the paper addressing all the comments raised carefully and satisfactorily.

According to all of the comments, we have revised the manuscript extensively and corrected several mistakes in our previous draft. If there are any other modifications we could make, we would like very much to modify them and we really appreciate your help. We hope that the changes we have made resolve all your concerns about the article. We are more than happy to make any further changes that will improve the paper or facilitate successful publication.

Responds to Reviewer’s comments:

1. If the authors have adequately addressed your comments raised in a previous round of review and you feel that this manuscript is now acceptable for publication, you may indicate that here to bypass the “Comments to the Author” section, enter your conflict of interest statement in the “Confidential to Editor” section, and submit your "Accept" recommendation.

Reviewer #3: (No Response)

Thank you for reviewing our manuscript.

2. Is the manuscript technically sound, and do the data support the conclusions?

Reviewer #3: Yes

Thank you for your positive comments and valuable suggestions to improve the quality of our manuscript.

3. Has the statistical analysis been performed appropriately and rigorously?

Reviewer #3: Yes

Thank you for your positive comments and valuable suggestions to improve the quality of our manuscript.

4. Have the authors made all data underlying the findings in their manuscript fully available?

Reviewer #3: Yes

Thank you for your positive comments and valuable suggestions to improve the quality of our manuscript.

5. Is the manuscript presented in an intelligible fashion and written in standard English?

Reviewer #3: Yes

Thank you again for your positive comments and valuable suggestions to improve the quality of our manuscript.

6. Review Comments to the Author

Reviewer #3: This paper uses a two-stage Data Envelopment Analysis (DEA) model to calculate the production efficiency of China's real estate industry across 31 regions in 2014-2022. It uses a panel Tobit model to identify factors (the leverage ratio of companies, per capita GDP and regulatory policies) that significantly impact efficiency.

In Eq. 2, the methodology uses Tobit regression to explain efficiency. There are other regression methods such as truncated regression, robust-OLS regression or Papke-Wooldridge which are probably more suited method for these purposes. Adding a short discussion of these alternative ways for this type of analysis will be helpful (see Simar and Wilson 2008, Banker and Natarajan 2007 for discussion).

Your recommendations have significantly enhanced the rigor of this article. After thoroughly examining the three articles you recommended, we now possess a more comprehensive understanding of econometric analysis methods for determining the influencing factors of restricted dependent variables, such as truncated regression. Consequently, we have made corresponding modifications to the paper. Additionally, we faithfully included the article you recommended in our list of the references list. Once again, thank you for your meticulous review.

The following are the corresponding parts of the text:

After assessing the production efficiency of the real estate industry in different regions, this paper considers the efficiency values as dependent variables, and factors other than input-output data, reflecting the heterogeneity of the decision-making units as well as the regions, as independent variables. Drawing on insights from Amemiya, T.[28], Simar, L., and Wilson, P. W.[29], Banker, R. D., and Natarajan, R. regarding the analysis of influencing factors of restricted dependent variables, it is noted that "two-stage DEA-based procedures with OLS, ML, or even Tobit estimation in the second stage significantly outperform parametric methods"[30]. Given the practicality of computational software usage, this article employs a Tobit model to analyze the influencing factors affecting real estate production efficiency at this stage. The basic form of the model is presented in equation (2). 

7. PLOS authors have the option to publish the peer review history of their article (what does this mean?). If published, this will include your full peer review and any attached files.

Do you want your identity to be public for this peer review? For information about this choice, including consent withdrawal, please see our Privacy Policy.

Reviewer #3: No

Thank you for reviewing our manuscript.

---

## [Editor Report · Decision Letter 2]

16 Sep 2024

An   Analysis of Production Efficiency in China's Real Estate Industry Based on a Two-Stage DEA Model

PONE-D-23-38796R2

Dear Dr. Meng,

We’re pleased to inform you that your manuscript has been judged scientifically suitable for publication and will be formally accepted for publication once it meets all outstanding technical requirements.

Kind regards,

Ahmad Hassan Ahmad

Academic Editor

PLOS ONE

Additional Editor Comments (optional):

As you have adequately addressed the concerns raised by the reviewer, I am happy to accept the paper for publication. However, you need to follow the journal's specific instructions.
---

## [Editor Report · Acceptance letter]

11 Oct 2024

PONE-D-23-38796R2 

PLOS ONE

Dear Dr. Meng, 

I'm pleased to inform you that your manuscript has been deemed suitable for publication in PLOS ONE. Congratulations! Your manuscript is now being handed over to our production team.

Kind regards, 

on behalf of

Dr. Ahmad Hassan Ahmad 

Academic Editor

PLOS ONE